

# Scalable neighbour search and alignment with uvaia

Leonardo de Oliveira Martins[1], Alison E. Mather[1,2] and Andrew J. Page[1]

[1] Quadram Institute Bioscience, Norwich, United Kingdom
[2] University of East Anglia, Norwich, United Kingdom

## ABSTRACT

Despite millions of SARS-CoV-2 genomes being sequenced and shared globally, manipulating such data sets is still challenging, especially selecting sequences for focused phylogenetic analysis. We present a novel method, uvaia, which is based on partial and exact sequence similarity for quickly extracting database sequences similar to query sequences of interest. Many SARS-CoV-2 phylogenetic analyses rely on very low numbers of ambiguous sites as a measure of quality since ambiguous sites do not contribute to single nucleotide polymorphism (SNP) differences. Uvaia overcomes this limitation by using measures of sequence similarity which consider partially ambiguous sites, allowing for more ambiguous sequences to be included in the analysis if needed. Such fine-grained definition of similarity allows not only for better phylogenetic analyses, but could also lead to improved classification and biogeographical inferences. Uvaia works natively with compressed files, can use multiple cores and efficiently utilises memory, being able to analyse large data sets on a standard desktop.

# INTRODUCTION

Genome sequencing has been globally deployed at pace to understand the evolution, transmission and dynamics of the SARS-CoV-2 virus, with the goal of providing actionable data for management of the COVID-19 pandemic (*Du Plessis et al., 2021*; *Maxmen, 2021*; *Lambrou et al., 2022*). Genomic epidemiology contextualises newly sequenced genomes within currently available knowledge, as to whether these new sequences have been observed before, and when and where their most similar genomes have been sequenced before. This can allow for outbreaks to be identified, linked, and mitigations put in place to monitor or limit further spread. This is particularly important for closed environments such as hospitals (*Page et al., 2021*), care homes (*Aggarwal et al., 2021*), or for limiting the spread of newly emergent variants with concerning mutations (*Aggarwal et al., 2022*).

The number of SARS-CoV-2 genomes available in global public databases such as the ENA/NCBI and GISAID has surpassed 14 million (accessed 2023-01-23). SARS-CoV-2 is now the most sequenced organism of all time; however, bioinformatics methods have struggled to keep pace with the scale of the data, or are optimised for different properties (*e.g.*, small numbers of large genomes, rather than large numbers of small genomes). To complicate things further, the sequencing methods commonly utilised for SARS-CoV-2

Corresponding author
Leonardo de Oliveira Martins,
Leonardo.de-Oliveira-
Martins@quadram.ac.uk

can result in partial genomes (*Baker et al., 2021*). This can be due to a low viral load of the sample where a patient is at an early or late stage of their infection (*Alikhan et al., 2021*), due to a mutation causing a dropout in an amplicon primer sequence (*Sanderson & Barrett, 2021*), or could be due to the manner in which the sample has been collected and stored (*Liu et al., 2021*). On the other hand, laboratories might erroneously impute the reference DNA state to low coverage regions (*Baker et al., 2021*). Another algorithmic challenge is that SARS-CoV-2 mutates relatively slowly, with regular global lineage replacements as fitter lineages emerge. As circulating diversity can be very low during any given time period, any changes between sequences can be significant epidemiologically. To address huge genomic SARS-CoV-2 data sets with uneven quality, resolution, and completeness, uvaia uses similarity measures which indirectly account for genome completeness, as we'll see below.

To account for partial genomes, a minimum threshold for genome completeness is often applied, so that downstream analyses can rely on high quality genomes. For example, the COVID-19 Genomics UK (*COVID-19 Genomics UK Consortium, 2023*) consortium sets a threshold of 50% (*Page et al., 2021*), with data deposited in the ENA/NCBI and GISAID (>90% completeness). A cycle threshold (Ct), *i.e.,* the number of polymerase chain reaction amplification cycles necessary for the virus to be detected (*Rhoads et al., 2021*), is often applied before sequencing begins in order to maximise the chances of getting a high coverage genome. Thus higher Ct values mean that there is less viral RNA, and samples with Ct > 30 are usually excluded, to ensure that there is sufficient viral material available for sequencing. Phylogenetic analyses can be restricted to genomes with a higher completeness threshold, *e.g.,* at least 90% of the sites with high-coverage. This can result in clinically important samples being disregarded before sequencing, and for sequenced genomes to never be made public, even though they could contain epidemiologically useful information. Assuming that the sequence completeness does not compromise the alignment (*i.e.,* the presence of ambiguous sites does not generate a spurious alignment with regards to the reference genome), uvaia can find similar sequences based on the number of partial and exact matches, instead of the more common number of single nucleotide polymorphism (SNP) differences.

Two important software tools for the phylogenetic analysis of SARS-CoV-2 data sets are Nextstrain (*Hadfield et al., 2018*) and civet (*O'Toole et al., 2021*), and both contain strategies for reducing the number of sequences to those relevant to a particular study. Nextstrain can generate "focal sets" by downsampling based on geography and collection date information, which can subsequently be selected for inclusion through a genetic distance-based "priority" ordering (*Hodcroft et al., 2021*). Likewise, given a background database composed of alignment and metadata, civet can generate a "catchment area" composed of sequences within a given distance to the query genomes, with the possibility of downsampling (*O'Toole et al., 2021*). Both methods rely on stochastic factors, metadata information and a SNP-distance measure.

Another essential software for SARS-CoV-2 analysis with integrated phylogenetics is UShER, which can parsimoniously place a query sequence into a tree, and therefore can return the closest samples from this so-called mutation-annotated tree (*Turakhia et al.,*

*2021*). UShER relies on a given parsimonious tree, which can be further optimised and quickly updated with new sequences (*Ye et al., 2021*). Its native input format for sequences is the VCF format. However, it does not consider indels or ambiguous bases. UShER has good accuracy for PANGO lineage classification, and scales very well given an existing mutation-annotated tree (*Kramer et al., 2023*).

We have created a method, uvaia, for performing neighbourhood search of aligned sequences, allowing for similar genomes to be found within massive datasets even for more ambiguous sequences. Given a genome aligned to a reference, we can rapidly find all other similar, equivalent or identical genomes from massive public aligned data sets, and return a similarity matrix between them. Uvaia scales linearly to the massive datasets seen with SARS-CoV-2 genomics, and almost linearly per core to the query sample size. It accounts for the complexity of partial genomes, and can provide analysis rapidly on an ordinary laptop. Since it relies on aligned sequences, we also offer a reference-based aligner, uvaialign, as a convenient fallback which works readily with uvaia.

Uvaia has been used for rapidly analysing genomic data against massive databases to assist pandemic management in multiple countries, including distance based analyses analysing dynamics of the rapid emergence of the Omicron variant of concern in the UK (*Eales et al., 2022a*; *Eales et al., 2022b*), for phylogenetic based analyses to understand multiple waves in Zimbabwe (*Mashe et al., 2021a*; *Mashe et al., 2021b*), the spread of variants of concern in Pakistan (*Sarwar et al., 2021*), and the emergence and replacement of multiple variants of concern in Lebanon (*Merhi et al., 2022*). Uvaia achieves this by utilising high efficiency compression, efficient parallelisation and combines this with knowledge of the fundamental characteristics and properties found with SARS-CoV-2 genomic datasets. Uvaia is available under the open source GNU GPL3 licence from https://github.com/quadram-institute-bioscience/uvaia.

Uvaia addresses two problems related to massive data sets, which have not been fully explored by existing tools: by working natively with XZ-compressed files, and with a pool of sequences in parallel. The first is to avoid files which may fill up the disk space: the raw GISAID fasta file with all sequences as of v.2023_06_08, with more than 15Mi sequences, occupies 433.3 GiB, while its XZ-compressed equivalent takes only 1,618 MiB. Such files can also not be read into memory at once in most personal computers, and thus the uvaia programs work with the sequence files in manageable batches, using multiprocessing whenever possible, including the output XZ compression which is done in parallel.

## Handling ambiguity

Most SARS-CoV-2 sequences will have some indels but also a considerable number of Ns, which represent complete uncertainty or ambiguity in the base at the location. A sequence may also have partially ambiguous sites (IUPAC codes), as for instance the character M means that the site may be an A or a C, but not a G or a T. Both gaps and Ns are excluded from pairwise comparisons by most sequence comparison algorithms, including uvaia.

Uvaia calculates the pairwise similarity based on three measures: on unambiguous pairwise comparisons (*i.e.,* sites exclusively with A, C, G, or T on both sequences), on partial matches (so that an M will match A or C, for instance), and also on exact text

```
seq1  AAC GTT A--      7 valid sites: 7 x ACGT + 0 partial
seq2  AAC NNN AM-      5 valid sites: 4 x ACGT + 1 partial (M)
seq3  MNC GTT MC-      7 valid sites: 5 x ACGT + 2 partial (M)

ACGT_matches (seq1,seq2) = 4  partial_matches (seq1,seq2) = 4  valid_pair_comparisons(seq1,seq2) = 4
ACGT_matches (seq1,seq3) = 4  partial_matches (seq1,seq3) = 6  valid_pair_comparisons(seq1,seq3) = 6
ACGT_matches (seq2,seq3) = 1  partial_matches (seq2,seq3) = 4  valid_pair_comparisons(seq2,seq3) = 4
```

**Figure 1  Example of three sequences with no dissimilarities to each other (zero distance) which nonetheless contain differences.**

matches (such that an M matches with another M but not with A or C). The partial matches similarity is related to the polymorphism p-distance, which assumes that state ambiguity comes from population diversity (*Potts, Hedderson & Grimm, 2014*; *Zhao, Nielsen & Korneliussen, 2022*) .There is an option for uvaia to mimic other software by excluding partial matches from the comparison—although we notice that phylogenetic inference methods, particularly probabilistic, benefit from partially ambiguous information (*Felsenstein, 2003*; *Yang, 2014*) and thus are enabled by default in uvaia.

When we consider indels and Ns, looking at the distance between sequences may not be a good indicator of neighbourhood since the sequences may have few comparable sites (pairwise comparisons exclude sites with a gap or N in one of the sequences, see Fig. 1 for an example). The same caveat applies to percentage identity calculation, since we normalise by valid pairwise comparisons. Thus our "neighbourhood" (groups of similar genomic sequences) is defined by the total number of matches, instead of number of mismatches or fraction of matches.

## METHODS

Given a set of query sequences, we want to keep from a (potentially very large) reference alignment only the sequences which are close to at least one query. This keeps downstream inferences computationally feasible, and also helps faster inferences to be made, like neighbour-based lineage classification, or geographical analyses (*Sarwar et al., 2021*; *Eales et al., 2022a*). In uvaia a priority queue is created to store the neighbourhood of each query sample, where at most $k$ reference sequences are kept. For each query, a new reference is added to its queue, in order of importance, by its total number of unambiguous matches, of exact text matches, and of partial matches. Ties are further broken by the number of derived unambiguous matches, and ultimately by the number of valid sites of the reference sequence. The reference sequences are thus ranked for each query according to the tie-breaking statistics described above. This ranking therefore gives more weight to sequences with more unambiguous sites, and to comparisons with more textual matches (which in the worst case could be an artefact of the sequencing centre). The number of derived matches is based on the strict consensus between all queries, which is created to speed up calculations and split the sites into constant and polymorphic. The total number of matches is then the number of matches between the reference and the consensus across queries (*i.e.,* over constant sites amongst query sequences) plus the number of derived matches (calculated over polymorphic sites, that is which may differ between queries). The

rationale is to give preference to neighbours closer to the tips and farther from the ancestor of the query sequences. Portions of this text were previously published as part of a preprint (https://www.biorxiv.org/content/10.1101/2023.01.31.526458v2).

The same reference can be on the neighbourhood of more than one query sequence, but to speed up computations and to minimise duplicated effort, uvaia can remove identical and redundant query sequences. A sequence is redundant if there is another, more resolved sequence, with all its information but with less ambiguity. For example the 6-mer AACNNN is redundant with respect to AACAAA since the latter is a more resolved version of the former. Any close neighbour to the more resolved sequence will also be a close neighbour to the less resolved one. Notice that on the other hand NACAAA is not a more resolved version of AACNNN since there is information in the latter (the first "A") not available in the more resolved, former sequence.

Finally, a table with match information between each query and its set of N closest neighbouring reference sequences is output, together with a reduced reference alignment output with all temporary close neighbours. This reduced alignment file can be queried afterwards using the table information, to obtain the neighbour sequence themselves. Uvaia can also simulate other SNP distance algorithms where partially ambiguous sites are not taken into account (considering only A, C, G, and T).

The main program is thus uvaia, which uses the match similarities described above to, given a query data set of aligned sequences, rank and to extract the closest neighbours in a large reference data set. In addition to the main uvaia program, we also provide an implementation of the wavefront alignment algorithm (*Marco-Sola et al., 2020*) into a reference-based aligner, called uvaialign. Currently there are faster alternatives designed specifically for SAR-CoV-2 alignment (*Moshiri, 2021*; *Aksamentov et al., 2021*), but we maintain uvaialign for reproducibility (and ease of usage, in our opinion). This program is also multithreaded, works in batches to avoid exhausting available memory, and can read from and write to compressed format. It produces output which can be used by uvaia seamlessly, and was also designed to handle huge data sets. Uvaia and uvaialign have been used in several SARS-CoV-2 analyses already, some of which comprised all millions of sequences available at the time (*Sarwar et al., 2021*; *Eales et al., 2022a*; *Asante et al., 2023*), showing its scalability.

## RESULTS

Uvaia can also be used to calculate exhaustively the pairwise similarities between two alignment data sets, and we compared its results to those obtained using snp-dists (*Seemann, 2018*). Our test data set is composed of 6,000 unique sequences generated at the QIB (Norwich, UK) as part of the COG-UK consortium (*The COVID-19 Genomics UK consortium, 2020*), spanning the time range 2020–2022, and with the fraction of fully ambiguous sites (*i.e.,* Ns) between 0 and 50%. This data set was generated as follows: after downloading all sequences from COG-UK ("COG-UK archival version of data sets"), we selected all 33,488 samples sequenced at the QIB (labelled "NORW"), including those missing from public repositories due to high ambiguity (*Baker et al., 2021*). We removed

**Table 1 PANGO lineages (given by the "scorpio call" column) of the 6,000 unique sequences used in this study, divided into two data sets.**

| 1,000 samples data set: | 5,000 samples data set: |
| --- | --- |
| 341 Omicron (BA.1-like) | 1725 Omicron (BA.1-like) |
| 205 Delta (AY.4-like) | 1096 Delta (AY.4-like) |
| 196 Probable Omicron (BA.1-like) | 997 Probable Omicron (BA.1-like) |
| 145 Delta (B.1.617.2-like) | 719 Delta (B.1.617.2-like) |
| 55 Other | 249 Other |
| 20 Delta (AY.4.2-like) | 98 Alpha (B.1.1.7-like) |
| 18 Alpha (B.1.1.7-like) | 70 Delta (AY.4.2-like) |
| 10 Omicron (BA.2-like) | 26 Omicron (BA.2-like) |
| 4 Probable Omicron (BA.2-like) | 10 Probable Omicron (BA.3-like) |
| 3 Probable Omicron (Unassigned) | 7 Probable Omicron (BA.2-like) |
| 3 Probable Omicron (BA.3-like) | 1 Probable Omicron (Unassigned) |
| | 1 Gamma (P.1-like) |
| | 1 B.1.617.1-like |

completely identical ones using a 128-bit xxhash algorithm, and also removed those with more than 50% of fully ambiguous sites (since highly ambiguous regions compromise the alignment). All sequences were aligned with uvaialign against the isolate Wuhan-Hu-1 (Genbank accession no. MN908947.3) since both snp-dists and uvaia rely on aligned sequences. This test data set was further split into one set of 1,000 samples and one of 5,000 samples. Table 1 shows the distribution of PANGO lineages (*Rambaut et al., 2020*) of both sets to give an idea of their diversity. The 1,000 samples data set was then used to compare uvaia with distances calculated with snp-dists between all pairs.

In uvaia we calculate three similarity measures, based on the total number of ACGT matches, total number of text matches (*i.e.,* ACGT plus partially ambiguous treated as characters), and total number of partial matches (where the IUPAC ambiguity code is used to check for compatibility, except for Ns). Thus we can extract the number of mismatches by subtracting the number of pairwise comparisons by these similarity values. In Fig. 1 we show cases where they are not equivalent to the SNP distance, since usually these SNP calculations ignore all non-ACGT characters. Therefore even "identical" sequences as reported by *e.g.,* snp-dists may be quite distinct once we look at partially-ambiguous sites. In Fig. 2 we have density plots showing this difference for all pairwise comparisons using our smaller data set of 1,000 sequences. For every sequence pair, we calculate both their SNP distance (*i.e.,* number of SNPs which are different), and their "ACGT mismatch" as the difference between the number of valid comparisons (*i.e.,* sites where at least one is not a gap or N) and the number of ACGT matches. This is an indirect measure of the number of partially ambiguous sites. Figure 2 shows, then, the distribution of ACGT mismatches conditioned on the SNP distance. We see that even for pairs without any SNP difference (lowermost row in figure) most pairs had at least one partially ambiguous site between them. Therefore, especially for large databases where the number of sequences without

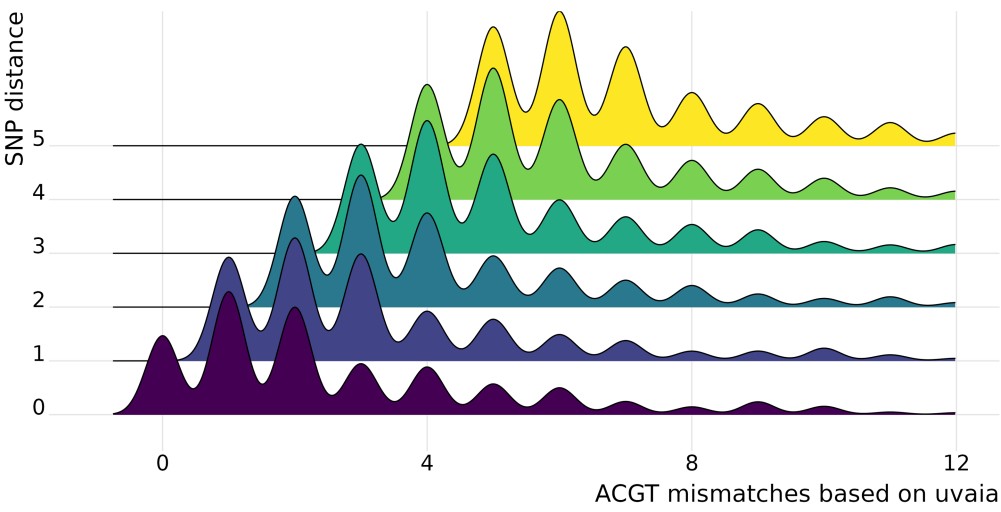

**Figure 2** **Stacked density plots (ridgeline plots).** Stacked density plots showing the difference between the pairwise sequence distance as calculated by snp-dists ($y$ axis) and the number of ACGT mismatches as the difference between the number of valid pairwise comparisons and the number of unambiguous DNA matches ($x$ axis). The $x$ axis is truncated at the 97% percentile to ease visualisation. We used a set of 1,000 sequences, such that 499,500 pairwise comparisons were performed and just those with a small number of differences are shown. Each density plot shows the distribution of ACGT mismatches for sequence pairs with the same SNP distance.

SNPs may be overwhelming, we need higher resolution exclusion criteria for neighbours search.

We performed a more thorough analysis of this difference for all pairwise comparisons, as well as using the other similarity measures (Text S1 and Fig. S3). We can then see that in fact it is more common to have at least one disagreement between uvaia-based and SNP-based distances than for the two measures to agree.

Uvaia can replicate snp-dists results by excluding partially ambiguous states, *i.e.*, treating them together with Ns and gaps. We then used uvaia to calculate the similarity between these 1,000 "query" sequences to 5,000 distinct "reference" sequences with similar distribution of lineages and ambiguous sites (using the partitioning of the 6,000 samples described previously). To show the effect of uncertainty, we compare the number of neighbours with no SNPs to the reference sequences to their number of partially unambiguous sites in Fig. 3. We see how less resolved sequences appear to have more "identical" neighbours, *i.e.*, sequences with no SNP differences. Thus, by using the number of matches instead of SNP differences, we can account for this ambiguity. Notice that here we used uvaia to calculate the SNP distance since snp-dists cannot, at the moment, calculate pairwise distances between two distinct sets, and furthermore uvaia already outputs the number of valid sites.

To observe the effect of sequences with partially ambiguous sites in a phylogenetic context, we inferred a maximum likelihood tree of closest neighbour sequences to an arbitrary sequence by uvaia or by UShER (Text S1 and Figs. S1–S2). The resulting phylogeny
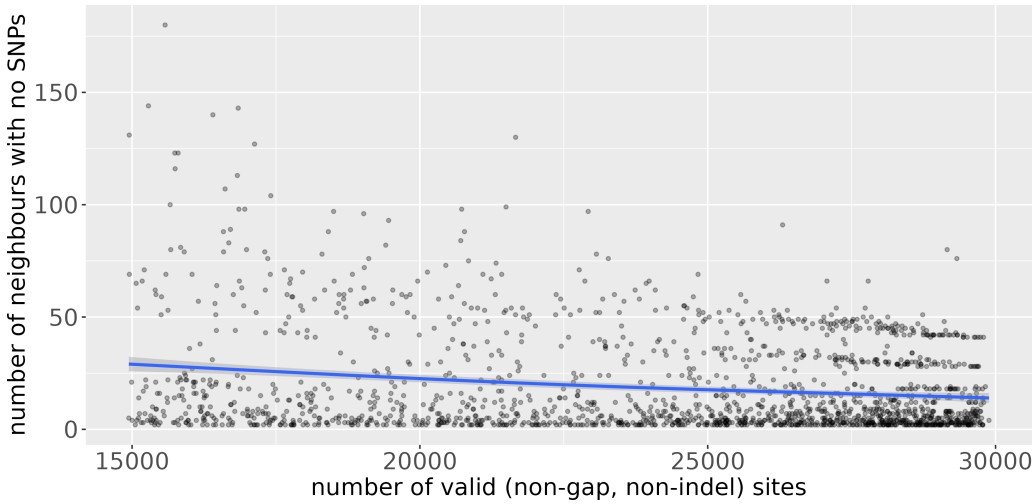

**Figure 3** **Scatterplot of number of "query" neighbours with no SNPs to "reference" sequences (*y* axis) against number of partially or completely unambiguous sites in the "reference" sequence (*x* axis).** The blue line represents the regression line smoothed with a generalised additive model (*Wood, 2017*). The lower the number of valid sites, the less "resolved" the reference sequence is, and we observe a tendency for a higher number of query sequences with no apparent SNP differences. The horizontal stripes of dots at the more resolved quadrant are possibly due to single clusters composed of queries and references.

(Fig. S1) shows that: (1) the likelihood algorithm implemented in RAxML-NG (*Kozlov et al., 2019*) was capable of using the partial information from some sequences to estimate a more resolved phylogeny; and that (2) SNP distance-based neighbours might miss such sequences.

While the execution time of uvaia increases linearly with the number of sequences in the reference database, the memory requirements are bounded by the number of query sequences, such that as long as all queries can fit into memory, uvaia can process arbitrarily large reference databases (Text S1 and Figs. S4–S5). There is some overhead associated with working with compressed files for both input and output (to benefit from the large gain in space savings offered by XZ), which may explain why uvaia is slightly slower than snp-dists, especially for small data sets (Text S1 and Fig. S6).

## DISCUSSION

For poorly-resolved sequences, the likelihood of encountering "identical" neighbours is higher. By "identical", we mean cases where there are no Single Nucleotide Polymorphisms (SNPs) or in other words when there is 100% identity among the comparable sites. Therefore we must consider these cases also in the comparison, by (1) restricting our analysis to well-resolved sequences (*i.e.,* where the fraction of unambiguous sites is negligible) or by (2) looking at the matches instead of mismatches, focusing on the similarity instead of the distance, while accounting for the number of partially unambiguous sites. Most research in SARS-CoV2 has employed strategy (1), by including only sequences with *e.g.,*

90% unambiguous sites. Here we provide a software to solve (2), which can return the reference sequences with more matches to a query sequence.

This may affect not only the phylogenetic inference—since likelihood and Bayesian methods can use the ambiguity information (*Felsenstein, 2003*; *Yang, 2014*)—but also in other epidemiological analyses where finding the closest sequences may be challenging. Uvaia has been used successfully in both cases, by improving contact tracing and travel history inferences, and overall evolutionary analyses, and here we present further details into the algorithm.

Some pipelines might inadvertently replace ambiguous sites by the reference allele, mistakenly assuming that low coverage sites are evidence for no change (the ancestral state). This imputation strategy may inadvertently lead to misleading inferences since, as indicated by Fig. 3, it can negatively affect similarity and distance estimates. Furthermore it can mask the effects of recombination, intra-population diversity, homoplasies, and can mislead phylogenetic analyses (*Baker et al., 2021*). All similarity measures may be affected by such artefacts, and in some cases one measure may be more relevant than another. Our suggestion is to use more than one when selecting neighbours, as well as for instance the number of partial mismatches (see phylogenetic example in Text S1).

## CONCLUSION

We show how a SNP distance-based neighbour sequence search may have low resolution to find the most similar sequences in large databases. Notice that the main differences between uvaia and other algorithms is the inclusion of partially ambiguous sites (with the possibility of incorporating or not their compatibility information), and the number of matching sites as optimality criterion. It is known that such partial matches can affect phylogenetic analyses, with a few evolutionary distances incorporating partially ambiguous sites as informative characters (*Potts, Hedderson & Grimm, 2014*; *Joly, Bryant & Lockhart, 2015*), and a so-called "Intra-Individual Site Polymorphism" is available in the R library phangorn (*Schliep, 2011*). These distances fare well in comparison to others (*Zhao, Nielsen & Korneliussen, 2022*), and have been used in phylogenetic studies (*Scheunert & Heubl, 2017*). We also have shown before (*Baker et al., 2021*) how in SARS-CoV-2 the sequencing (and bioinformatic) handling of partially ambiguous sites can lead to differences in the resulting phylogeny under maximum likelihood. As we show here, including these partially ambiguous sites changes what we consider as "valid" comparisons and can increase the resolution of a sequence neighbourhood.

We note that working with compressed files has a computational cost, even when using a multithreaded algorithm for compression; decompression is always single threaded, but faster than compression. The advantages of uvaia are best seen in a restricted budget environment, with finite disk and memory resources, but fully using multicore architectures available even in lower end laptops. And as we are already seeing for SARS-CoV-2, in preparation for future pandemics, new software like uvaia will be the default, scalable not for thousands but for millions of sequences.

## ACKNOWLEDGEMENTS

The authors would like to thank the Norfolk and Norwich University Hospitals NHS Foundation trust for providing samples for sequencing, and the Bioinformatics and Sequencing teams at the Quadram for generating all data analysed in the present study.

### Funding

This research was funded by the Biological Sciences Research Council (BBSRC) Institute Strategic Programme Microbes in the Food Chain BB/R012504/1 and its constituent project BBS/E/F/000PR10348 (Theme 1, Epidemiology and Evolution of Pathogens in the Food Chain), also Quadram Institute Bioscience BBSRC funded Core Capability Grant (project number BB/CCG1860/1). The funders had no role in study design, data collection and analysis, decision to publish, or preparation of the manuscript.

### Grant Disclosures

The following grant information was disclosed by the authors:
Biological Sciences Research Council (BBSRC) Institute Strategic Programme Microbes in the Food Chain: BB/R012504/1.
Theme 1, Epidemiology and Evolution of Pathogens in the Food Chain: BB-S/E/F/000PR10348.
Quadram Institute Bioscience BBSRC funded Core Capability Grant: BB/CCG1860/1.

### Competing Interests

The authors declare there are no competing interests.

### Author Contributions

- Leonardo de Oliveira Martins conceived and designed the experiments, performed the experiments, analyzed the data, prepared figures and/or tables, authored or reviewed drafts of the article, and approved the final draft.
- Alison E. Mather conceived and designed the experiments, authored or reviewed drafts of the article, and approved the final draft.
- Andrew J. Page conceived and designed the experiments, authored or reviewed drafts of the article, and approved the final draft.

### Data Availability

Sequence read data and consensus genomes from COGUK are available at the European Nucleotide Archive: PRJEB37886.

All data are also available at GitHub: https://github.com/quadram-institute-bioscience/uvaia.

Video tutorials are available at Youtube: https://www.youtube.com/playlist?list=PLPZ2aSS2ApqoU6-FCd2H035uJHnLu9fTL.

## Supplemental Information

Supplemental information for this article can be found online at http://dx.doi.org/10.7717/peerj.16890#supplemental-information.

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
