# Peer review of "Scalable neighbour search and alignment with uvaia"

_PeerJ, doi:10.7717/peerj.16890_

## Round 0.1 · original submission · Major Revisions

Please address the concerns of the reviewers and revise the manuscript accordingly.

·

Basic reporting

While in general the paper is well written, there are several issues with English language use (I note some of them below). The Conclusion is unfortunately poorly written and currently contains material which should be in the Discussion.

Languages issues noted:

l. 95 - "compromise the" - exhaust
l. 97 - "Such files cannot be read into memory at once in most personal computers either" - "Such files can also not be read into memory at once on most personal computers"
l. 113 - "phylogenetic inference methods, specially probabilistic ones, do benefit from partially ambiguous information and thus are also used by default in uvaia" - "phylogenetic inference methods, especially probabilistic ones, benefit from partially ambiguous information and thus partial matches are enabled by default in uviava"
- cite an example of a method using this information
l. 127 - delete "In particular"
l. 147 - "At the end" - "finally"
l. 150 - delete , after information,
l. 153 - "The main program is thus uvaia, which uses the match similarities described above to, given a query data set of aligned sequences, ranks and extracts the closest neighbours in a large reference data set. We also have incorporated an implementation of the wavefront alignment " - "In addition to the main uvaia program, we also provide an implementation of the wavefront aligment"
l. 157 - "works in batches to avoid using all available memory up" - "works in batches to avoid exausting memory"

Citation practices:

l. 210 - it is not necessary to keep repeating the papers in which this software was used.

Experimental design

The methods were well described, but the comparison with snp-dists suffered some weaknesses:

1. On l. 168 reference is made to "this set" - I presume, but am not sure, that this means the smaller set (the one of 1000 sequences) mentioned previously.

2. The uvaialign tool is used as mentioned in line 166. Once again, I presume that this was to allow straightforward comparison with snp-dists?

3. I find Figure 2 difficult to read. Conventionally box and whisker plots have solid horizontal lines at the end of the "whisker" and don't have the dots shown above the end of the "whiskers" in this plot.

Validity of the findings

l. 162 - the details of the set are not sufficiently described. Supplementary Table 1 lists 1967 sequences, but the experiment mentions use of 6000 and a split of the set into 5000 and 1000. The data thus needs better description before this paper is considered ready for publication.

Reviewer 2 ·

Basic reporting

Needs some English work (see line comments below).

Line 14- This statement is unclear. What exactly is Uvaia alleviating? What role is Uvaia playing in these workflows? Retrieving similar sequences based on ambiguous sites is not helpful for a phylogenetic analysis relying on minimal ambiguity.
Line 25- This language is overly simplistic.
Line 37- cite the first clause (“low viral load”) to match the rest of the statement and other causes.
Line 41- runon sentence. “Making every mutation count” is vague and colloquial.
Line 44- delete “was created so that it only” as it does not add meaningful information.
Line 50- integrate the definition of CT more naturally into the statement without using a colon
Line 54- sentence fragment. Decide what?
Line 62- Nextstrain is misspelled
Line 74- rather than saying it is dependent on VCF, it is more correct to say that it depends on an alignment procedure. Aligned sequences can be converted to VCF or Diff formats ingested by UShER. Uvaia relies on aligned sequences in exactly the same way. Additionally “relying on the existence of the maximum parsimony tree” is a vague statement, somehow implying that the tree might not exist- it certainly exists, though it may be an inaccurate representation of some relationships. Reword this statement. Additionally, what benefits does Uvaia have as an alternative to UShER, or in combination with it?
Line 94- check grammar (remove colon, “tries to solve two problems related to massive datasets… by working natively…)
Line 105- comparison -> comparisons
Line 112- populational -> population
Line 120- Figure is missing from the review manuscript.
Line 136- clarify this statement.
Line 154 - revise for grammar (“above to… ranks”->”above to… rank”, etc)
Line 169- repetitive, rephrase
Line 215- this is a valid concern, but feels dropped in the discussion largely without context. Consider moving this to a location where the nature of ambiguous sequence is discussed.
Line 231- what time benefit? Reference explicit benchmarking material.

Experimental design

No comment.

Validity of the findings

No comment.

Additional comments

The major benefit to Uvaia over gold standard options appears to be that it can retrieve and handle partially ambiguous sequences more effectively. However, it is not explicitly discussed what purpose retrieving ambiguous or low quality sequences serves. The authors should consider including case studies or more elaborated explanations for effective use cases for Uvaia and ambiguous sequences.

The uvaialign tool presented to create the reference database is of significant interest, both on its own merits and in the role it plays in the use of Uvaia, but receives relatively little attention in this manuscript. The authors should elaborate further on this implementation and how it compares to Minimap2 and other standard alignment tools for SARS-CoV-2.

The Uvaia examples presented in the manuscript and available through Github only use about a few thousand sequences in the reference database. This is multiple orders of magnitude smaller than the millions of sequences available, and since Uvaia is explicitly attempting to address the scale of the pandemic, the authors should directly demonstrate its capabilities in working with these large datasets. If it cannot, then they need to acknowledge that weakness. Accordingly, this manuscript would benefit from explicit runtime and memory use benchmarking, against both query and reference dataset sizes. If a user were to query the complete SARS-CoV-2 dataset of more than 10 million samples, how long will Uvaia take to return results, and is it actually possible on a standard laptop’s available memory?

Tools like UShER are capable of extracting neighborhoods for an arbitrary number of query sequences from a dataset of millions of sequences in minutes; how does Uvaia compare in practice to this? How does the size of the reference base and time of construction scale with available data? Additionally, UCSC provisions UShER MATs for public use, serving as a preconstructed database containing all available data. The authors should consider providing something similar for users to apply Uvaia to in querying their sequences of interest.

In terms of implementation, it appears that Uvaia is only available on Linux. This is in contradiction to one of the authors outlined key benefits, being scalability and portability of the approach to run on single laptops. Most laptop users are on MacOS or Windows; while Windows has the WSL subsystem, Uvaia would benefit substantially from having MacOS conda builds provisioned.

All figure captions could use some additional explanation. A boxplot is an inappropriate choice for Figure 2, especially given the invariability between the conditions. Figure 3 is better suited as a supplemental figure, given its very simple conclusion (no differences between text and ACGT mismatches) and conceptual overlap with Figure 2. In Figure 4, there is some structure with separate lines in the number of neighbors for highly complete sequences, that should be mentioned and explained in the figure caption.

Cite this review as

·

Basic reporting

This is a clearly written article with good structure. Raw data is shared, and code is available on GitHub.

L62 : Nextrain
L200: slightly awkward sentence (not clear what "It" refers to), could consider "It is more likely that [identical neighbours will be found]" or another rephrase

I came across the authors' YouTube videos about uvaia. These could be interesting materials to link from the paper.

Experimental design

I believe the requirements are satisfied.

If I understand the methods correctly, then if a query has an M [meaning "A or C"] at a position, then during ranking a reference with "M" will be preferred to a reference with "A" or with "C". I can imagine some downsides to this, as non-ambiguous sequences might be useful to help resolve the ambiguity. (But also advantages in some contexts).

It might have been interesting to report on the specifics of the performance provided by uvaia in terms of runtime and memory.

Validity of the findings

Code is available and good documentation is provided. I was able to install and run the software successfully, and it seemed of high quality.

"Such fine-grained definition of similarity allows not only for better phylogenetic analyses, but also for improved classification and biogeographical inferences" - while this is plausible, I guess it is not directly shown in the manuscript. I think there is a plausible argument that in rare cases consideration of ambiguity codes could actually create problems for analyses. My experience of the SARS-CoV-2 pandemic has been that sequencing pipelines differ substantially in their propensity to insert ambiguity codes. Given the point above about preferring "M" = "M" text-matches, this could end up specifically selecting neighbourhood sequences from the same sequencing centre, in an artifactual way.

---

## Round 0.2 · Minor Revisions

Please address the remaining minor concerns of the reviewer and revise the manuscript accordingly.

·

Basic reporting

The language is much improved and the article is straightforward to read

Experimental design

While there are other tools now doing similar things to uvaia, the motivation for the software and comparison with other software is clearly explained.

Validity of the findings

Sufficient data is provided to replicate the findings.

Reviewer 2 ·

Basic reporting

The manuscript has significantly improved in English quality and wording. However, I have a few additional suggestions, focused on the introductory section:

Line 37- replace the comma with a semicolon (e.g. …of all time; however, …)
Line 50- delete “uses the sequence data and”. It’s already implied that the sequence data is being used. Additionally, remove the dashes.
Line 54- algorithms are applied or implemented, not “focused”- reword.
Line 84- sentence is run on; divide into statements describing basic functionality and potential drawbacks (not considering indels or ambiguous bases)
Line 109- “tries to solve” -> “addresses”
Line 110- remove the repeated “by working” (e.g. …files and a pool of sequences in parallel)
Line 220- Remove the beginning “And”

Experimental design

No comment.

Validity of the findings

No comment.

Additional comments

I appreciate the detailed response to my previous review. I particularly appreciate that you've added examples for the application of ambiguous sequences for likelihood phylogenetics, and called out to this in the results.

I will add that while I understand that uvialign is not considered by the authors to be competitive with minimap2 and similar tools, it may be worth briefly elaborating that uvialign is designed to work directly with uvaia and serves as a convenient out-of-the-box option if nothing else in the introduction.

Uvaia clearly has its own niche in the this field. I believe this is a worthy publication at this stage. Congratulations on the excellent work!

Cite this review as

---

## Round 0.3 · accepted · Accept

All remaining concerns of the reviewer were adequately addressed and the revised manuscript is acceptable now.